# Functional Adversarial Attacks

**Cassidy Laidlaw**
University of Maryland
claidlaw@umd.edu

**Soheil Feizi**
University of Maryland
sfeizi@cs.umd.edu

## Abstract

We propose *functional adversarial attacks*, a novel class of threat models for crafting adversarial examples to fool machine learning models. Unlike a standard $\ell_p$-ball threat model, a functional adversarial threat model allows only a *single* function to be used to perturb input features to produce an adversarial example. For example, a functional adversarial attack applied on colors of an image can change *all* red pixels simultaneously to light red. Such global uniform changes in images can be less perceptible than perturbing pixels of the image individually. For simplicity, we refer to functional adversarial attacks on image colors as ReColorAdv, which is the main focus of our experiments. We show that functional threat models can be combined with existing additive ($\ell_p$) threat models to generate stronger threat models that allow both small, individual perturbations and large, uniform changes to an input. Moreover, we prove that such combinations encompass perturbations that would not be allowed in either constituent threat model. In practice, ReColorAdv can significantly reduce the accuracy of a ResNet-32 trained on CIFAR-10. Furthermore, to the best of our knowledge, combining ReColorAdv with other attacks leads to the strongest existing attack even after adversarial training.

## 1 Introduction

There is an extensive recent literature on *adversarial examples*, small perturbations to inputs of machine learning algorithms that cause the algorithms to report an erroneous output, e.g. the incorrect label for a classifier. Adversarial examples present serious security challenges for real-world systems like self-driving cars, since a change in the environment that is not noticeable to a human may cause unexpected, unwanted, or dangerous behavior. Many methods of generating adversarial examples (called *adversarial attacks*) have been proposed [23, 5, 15, 17, 3]. Defenses against such attacks have also been explored [18, 14, 30].

Most existing attack and defense methods consider a threat model of adversarial attacks where adversarial examples can differ from normal inputs by a small $\ell_p$ distance. However, using this threat model that encompasses a simple definition of "small perturbation" misses other types of perturbations that may also be imperceptible to humans. For instance, small spatial perturbations have been used to generate adversarial examples [4, 27, 26].

In this paper, we propose a new class of threat models for adversarial attacks, called *functional threat models*. Under a functional threat model, adversarial examples can be generated from a regular input to a classifier by applying a *single* function to all features of the input:

$$\text{Additive threat model:} \quad (x_1, \ldots, x_n) \quad \rightarrow \quad (x_1 + \delta_1, \ldots, x_n + \delta_n)$$
$$\textbf{Functional threat model:} \quad (x_1, \ldots, x_n) \quad \rightarrow \quad (f(x_1), \ldots, f(x_n))$$

For instance, the perturbation function $f(\cdot)$ could darken every red pixel in an image, or increase the volume of every timestep in an audio sample. Functional threat models are in some ways more restrictive because features cannot be perturbed individually. However, the uniformity of the

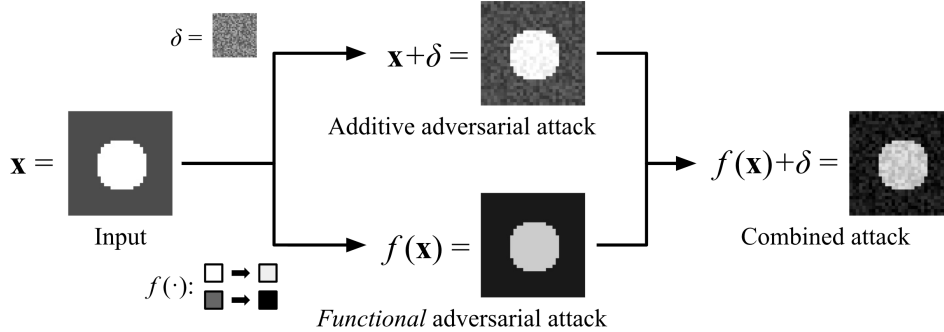

Figure 1: A visualization of an additive adversarial attack, a functional adversarial attack, and their combination. The additive attack perturbs each feature (pixel) separately, whereas the functional attack applies the same function $f(\cdot)$ to every feature.

perturbation in a functional threat model makes the change less perceptible, allowing for larger absolute modifications. For example, one could darken or lighten an entire image by quite a bit without the change becoming noticeable. This stands in contrast to separate changes to each pixel, which must be smaller to avoid becoming perceptible. We discuss various regularizations that can be applied to the perturbation function $f(\cdot)$ to ensure that even large changes are imperceptible.

The advantages and disadvantages of additive ($\ell_p$) and functional threat models complement each other; additive threat models allow small, individual changes to every feature of an input while functional threat models allow large, uniform changes. Thus, we combine the threat models (see figure 1) and show that the combination encompasses more potential perturbations than either one separately, as we explain in the following theorem which is stated more precisely in section 3.2.

**Theorem 1** (informal). *Let* $\mathbf{x}$ *be a grayscale image with* $n \geq 2$ *pixels. Consider an additive threat model that allows changing each pixel by up to a certain amount, and a functional threat model that allows darkening or lightening the entire image by a greater amount. Then the combination of these threat models allows potential perturbations that are not allowed in either constituent threat model.*

Functional threat models can be used in a variety of domains such as images (e.g. by uniformly changing image colors), speech/audio (e.g. by changing the "accent" of an audio clip), text (e.g. by replacing a word in the entire document with its synonym), or fraud analysis (e.g. by uniformly modifying an actor's financial activities). Moreover, because functional perturbations are large and uniform, they may also be easier to use for physical adversarial examples, where the small pixel-level changes created in additive perturbations could be drowned out by environmental noise.

In this paper, we will focus on one such domain—images—and define ReColorAdv, a functional adversarial attack on pixel colors (see figure 2). In ReColorAdv, we use a flexibly parameterized function $f$ to map each pixel color $c$ in the input to a new pixel color $f(c)$ in an adversarial example. We regularize $f(\cdot)$ both to ensure that no color is perturbed by more than a certain amount, and to make sure that the mapping is smooth, i.e. similar colors are perturbed in a similar way. We show that ReColorAdv can use colors defined in the standard red, green, blue (RGB) color space and also in CIELUV color space, which results in less perceptually different adversarial examples (see figure 4).

We experiment by attacking defended and undefended classifiers with ReColorAdv, by itself and in combination with other attacks. We find that ReColorAdv is a strong attack, reducing the accuracy of a ResNet-32 trained on CIFAR-10 to 3.0%. Combinations of ReColorAdv and other attacks are yet more powerful; one such combination lowers a CIFAR-10 classifier's accuracy to 3.6%, even after adversarial training. This is lower than the previous strongest attack of Jordan et al. [10]. We also demonstrate the fragility of adversarial defenses based on an additive threat model by reducing the accuracy of a classifier trained with TRADES [30] to 5.7%. Although one might attempt to mitigate the ReColorAdv attack by converting images to grayscale before classification, which removes color information, we show that this simply decreases a classifier's accuracy (both natural and adversarial). Furthermore, we find that combining ReColorAdv with other attacks improves the strength of the attack without increasing the perceptual difference, as measured by LPIPS [32], of the generated adversarial example.

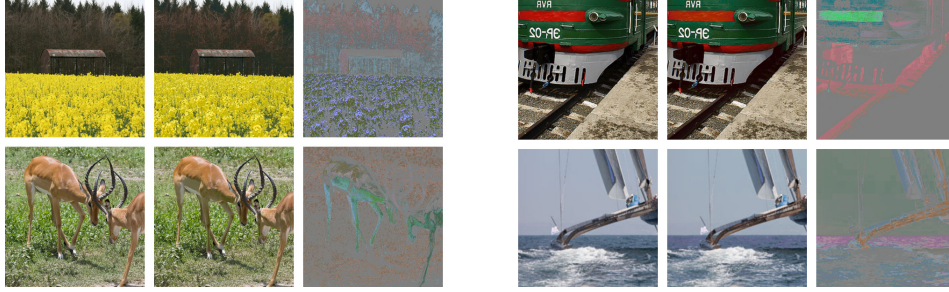

Figure 2: Four ImageNet adversarial examples generated by ReColorAdv against an Inception-v4 classifier. From left to right in each group: original image, adversarial example, magnified difference.

Our contributions are summarized as follows:

- We **introduce** a novel class of threat models, functional adversarial threat models, and combine them with existing threat models. We also describe ways of regularizing functional threat models to ensure that generated adversarial examples are imperceptible.

- **Theoretically**, we prove that additive and functional threat models combine to create a threat model that encompasses more potential perturbations than either threat model alone.

- **Experimentally**, we show that ReColorAdv, which uses a functional threat model on images, is a strong adversarial attack against image classifiers. To the best of our knowledge, combining ReColorAdv with other attacks leads to the strongest existing attack even after adversarial training.

## 2   Review of Existing Threat Models

In this section, we define the problem of generating an adversarial example and review existing adversarial threat models and attacks.

**Problem Definition**    Consider a classifier $g : \mathcal{X}^n \to \mathcal{Y}$ from a feature space $\mathcal{X}^n$ to a set of labels $\mathcal{Y}$. Given an input $\mathbf{x} \in \mathcal{X}^n$, an adversarial example is a slight perturbation $\widetilde{\mathbf{x}}$ of $\mathbf{x}$ such that $g(\widetilde{\mathbf{x}}) \neq g(\mathbf{x})$; that is, $\widetilde{\mathbf{x}}$ is given a different label than $\mathbf{x}$ by the classifier. Since the aim of an adversarial example is to be perceptually indistinguishable from a normal input, $\widetilde{\mathbf{x}}$ is usually constrained to be close to $\mathbf{x}$ by some *threat model*. Formally, Jordan et al. [10] define a threat model as a function $t : \mathcal{P}(\mathcal{X}^n) \to \mathcal{P}(\mathcal{X}^n)$, where $\mathcal{P}$ denotes the power set. The function $t(\cdot)$ maps a set of classifier inputs $\mathcal{S}$ to a set of perturbed inputs $t(\mathcal{S})$ that are imperceptibly different. With this definition, we can formalize the problem of generating an adversarial example from an input:

$$\text{find } \widetilde{\mathbf{x}} \quad \text{such that } g(\widetilde{\mathbf{x}}) \neq g(\mathbf{x}) \text{ and } \widetilde{\mathbf{x}} \in t(\{\mathbf{x}\})$$

**Additive Threat Model**    The most common threat model used when generating adversarial examples is the additive threat model. Let $\mathbf{x} = (x_1, \ldots, x_n)$, where each $x_i \in \mathcal{X}$ is a feature of $\mathbf{x}$. For instance, $x_i$ could correspond to a pixel in an image or the filterbank energies for a timestep in an audio sample. In an additive threat model, we assume $\widetilde{\mathbf{x}} = (x_1 + \delta_1, \ldots, x_n + \delta_n)$; that is, a value $\delta_i$ is added to each feature of $\mathbf{x}$ to generate the adversarial example $\widetilde{\mathbf{x}}$. Under this threat model, perceptual similarity is usually enforced by a bound on the norm of $\delta = (\delta_1, \ldots, \delta_n)$. Thus, the additive threat model is defined as

$$t_{\text{add}}(\mathcal{S}) \triangleq \{(x_1 + \delta_1, \ldots, x_n + \delta_n) \mid (x_1, \ldots, x_n) \in \mathcal{S}, \|\delta\| \leq \epsilon\}.$$

Commonly used norms include $\|\cdot\|_2$ (Euclidean distance), which constrains the sum of squares of the $\delta_i$, $\|\cdot\|_0$, which constrains the number of features can be changed, and $\|\cdot\|_\infty$, which allows changing each feature by up to a certain amount. Note that all of the $\delta_i$ can be modified individually to generate a misclassification, as long as the norm constraint is met. Thus, a small $\epsilon$ is usually necessary because otherwise the input could be made incomprehensible by noise.

Most previous work on generating adversarial examples has employed the additive threat model. This includes gradient-based methods like FGSM [5], DeepFool [15], and Carlini & Wagner [3], and gradient-free methods like SPSA [25] and the Boundary Attack [2].

**Other Threat Models**    Some recent work has focused on *spatial threat models*, which allow for slight perturbations of the locations of features in an input rather than perturbations of the features themselves [27, 26, 4]. Others have proposed threat models based on properties of a 3D renderer [29] and constructing adversarial examples with a GAN [22]. Finally, some research has focused on coloring-based threat models through modification of an image's hue and saturation [7], inverting images [8], using a colorization network [1], and applying an affine transformation to colors followed by PGD [31]. See appendix E for a discussion of non-additive threat models and comparison to our proposed functional threat model.

## 3    Functional Threat Model

In this section, we define *functional threat model* and explore its combinations with existing threat models. Recall that in the additive threat model, each feature of an input can only be perturbed by a small amount. Because all the features are changed separately, larger changes could make the input unrecognizable. Our key insight is that larger perturbations to an input should be possible if the dependencies between features are considered.

Unlike the additive threat model, in the functional threat model the features $x_i$ are transformed by a single function $f : \mathcal{X} \rightarrow \mathcal{X}$, called the perturbation function. That is,

$$\widetilde{\mathbf{x}} = f(\mathbf{x}) = (f(x_1), \ldots, f(x_n))$$

Under this threat model, features which have the same value in the input must be mapped to the same value in the adversarial example. Even large perturbations allowed by a functional threat model may be imperceptible to human eyes because they preserve dependencies between features (for example, shape boundaries and shading in images, see figure 1). Note that the features $x_i$ which are modified by the perturbation function $f(\cdot)$ need not be scalars; depending on the application, vector-valued features may be useful.

### 3.1    Regularizing Functional Threat Models

In the functional threat model, various regularizations can be used to ensure that the change remains imperceptible. In general, we can enforce that $f \in \mathcal{F}$; $\mathcal{F}$ is a family of allowed perturbation functions. For instance, we may want to bound by some small $\epsilon$ the maximum difference between the input and output of the perturbation function. In that case, we will have:

$$\mathcal{F}_{\text{diff}} \triangleq \{f : \mathcal{X} \rightarrow \mathcal{X} \mid \forall x_i \in \mathcal{X} \, \|f(x_i) - x_i\| \leq \epsilon\} \tag{1}$$

$\mathcal{F}_{\text{diff}}$ prevents absolute changes of more than a certain amount. Note that the $\epsilon$ bound may be higher than that of an additive model, since uniform changes are less perceptible. However, this regularization may not be enough to prevent noticeable changes. $\mathcal{F}_{\text{diff}}$ still includes functions that map similar (but not identical) features very differently. Therefore, a second constraint could be used that forces similar features to be perturbed similarly:

$$\mathcal{F}_{\text{smooth}} \triangleq \{f \mid \forall x_i, x_j \in \mathcal{X} \, \|x_i - x_j\| \leq r \Rightarrow \|(f(x_i) - x_i) - (f(x_j) - x_j)\| \leq \epsilon_{\text{smooth}}\} \tag{2}$$

$\mathcal{F}_{\text{smooth}}$ requires that similar features are perturbed in the same "direction". For instance, if green pixels in an image are lightened, then yellow-green pixels should be as well.

Depending on the application, these constraints or others may be needed to maintain an imperceptible change. We may want to choose $\mathcal{F}$ to be $\mathcal{F}_{\text{diff}}$, $\mathcal{F}_{\text{smooth}}$, $\mathcal{F}_{\text{diff}} \cap \mathcal{F}_{\text{smooth}}$, or an entirely different family of functions. Once we have chosen an $\mathcal{F}$, we can define a corresponding functional threat model as

$$t_{\text{func}}(\mathcal{S}) \triangleq \{(f(x_1), \ldots, f(x_n)) \mid (x_1, \ldots, x_n) \in \mathcal{S}, f \in \mathcal{F}\}$$

### 3.2    Combining Threat Models

Jordan et al. [10] argue that combining multiple threat models allows better approximation of the complete set of adversarial perturbations which are imperceptible. Here, we show that combining the additive threat model with a simple functional threat model can allow adversarial examples which are not allowable by either model on its own. The following theorem (proved in appendix A)

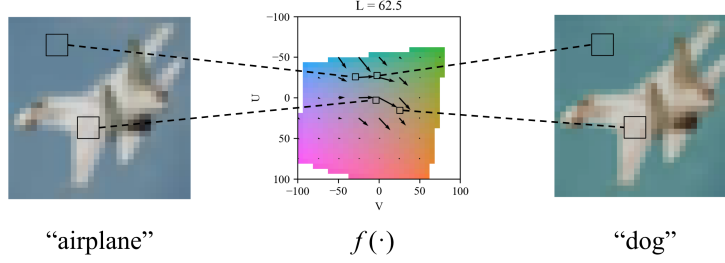

"airplane"                    $f(\cdot)$                    "dog"

Figure 3: ReColorAdv transforms each pixel in the input image $\mathbf{x}$ (left) by the same function $f(\cdot)$ (center) to produce an adversarial example $\widetilde{\mathbf{x}}$ (right). The perturbation function $f$ is shown as a vector field in CIELUV color space.

demonstrates this on images for a combination of an additive threat model which allows changing each pixel by a small, bounded amount and a functional threat model which allows darkening or lightening the entire image by up to a larger amount, both of which are arguably imperceptible transformations.

**Theorem 1.** *Let $\mathbf{x}$ be a grayscale image with $n \geq 2$ pixels, i.e. $\mathbf{x} \in [0,1]^n = \mathcal{X}^n$. Let $t_{add}$ be an additive threat model where the $\ell_\infty$ distance between input and adversarial example is bounded by $\epsilon_1$, i.e. $\|(\delta_1, \ldots, \delta_n)\|_\infty \leq \epsilon_1$. Let $t_{func}$ be a functional threat model where $f(x) = c\,x$ for some $c \in [1 - \epsilon_2, 1 + \epsilon_2]$ such that $\epsilon_2 > \epsilon_1 > 0$. Let $t_{combined} = t_{add} \circ t_{func}$. Then the combined threat model allows adversarial perturbations which are not allowed by either constituent threat model. Formally, if $\mathcal{S} \subseteq \mathcal{X}^n$ contains an image $\mathbf{x}$ that is not dark, that is $\exists x_i$ s.t. $x_i > \epsilon_1/\epsilon_2$, then*

$$t_{combined}(\mathcal{S}) \supsetneq t_{add}(\mathcal{S}) \cup t_{func}(\mathcal{S}) \qquad \text{or equivalently} \qquad \exists \widetilde{\mathbf{x}} \text{ s.t. } \begin{array}{l} \widetilde{\mathbf{x}} \in t_{combined}(\mathcal{S}) \\ \widetilde{\mathbf{x}} \notin t_{add}(\mathcal{S}) \cup t_{func}(\mathcal{S}) \end{array}$$

## 4  ReColorAdv: Functional Adversarial Attacks on Image Colors

In this section, we define ReColorAdv, a novel adversarial attack against image classifiers that leverages a functional threat model. ReColorAdv generates adversarial examples to fool image classifiers by uniformly changing colors of an input image. We treat each pixel $x_i$ in the input image $\mathbf{x}$ as a point in a 3-dimensional color space $\mathcal{C} \subseteq [0,1]^3$. For instance, $\mathcal{C}$ could be the normal RGB color space. In section 4.1, we discuss our use of alternative color spaces. We leverage a perturbation function $f : \mathcal{C} \to \mathcal{C}$ to produce the adversarial example. Specifically, each pixel in the output $\widetilde{\mathbf{x}}$ is perturbed from the input $\mathbf{x}$ by applying $f(\cdot)$ to the color in that pixel:

$$x_i = (c_{i,1}, c_{i,2}, c_{i,3}) \in \mathcal{C} \subseteq [0,1]^3 \quad \to \quad \widetilde{x}_i = (\widetilde{c}_{i,1}, \widetilde{c}_{i,2}, \widetilde{c}_{i,3}) = f(c_{i,1}, c_{i,2}, c_{i,3})$$

For the purposes of finding an $f(\cdot)$ that generates a successful adversarial example, we need a parameterization of the function that allows both flexibility and ease of computation. To accomplish this, we let $\mathcal{G} = g_1, \ldots, g_m \subseteq [0,1]^3$ be a discrete grid of points (or point lattice) where $f$ is explicitly defined. That is, we define parameters $\theta_1, \ldots, \theta_m$ and let $f(g_i) = \theta_i$. For points not on the grid, i.e. $x_i \notin \mathcal{G}$, we define $f(x_i)$ using trilinear interpolation. Trilinear interpolation considers the "cube" of the lattice points $g_j$ surrounding the argument $x_i$ and linearly interpolates the explicitly defined $\theta_j$ values at the 8 corners of this cube to calculate $f(x_i)$.

**Constraints on the perturbation function**   We enforce two constraints on $f(\cdot)$ to ensure that the crafted adversarial example is indistinguishable from the original image. These constraints are based on slight modifications of $\mathcal{F}_{\text{diff}}$ and $\mathcal{F}_{\text{smooth}}$ defined in section 3.1. First, we ensure that no pixel can be perturbed by more than a certain amount along each dimension in color space:

$$\mathcal{F}_{\text{diff-col}} \triangleq \{f : \mathcal{C} \to \mathcal{C} \mid \forall (c_1, c_2, c_3) \in \mathcal{G} \quad |c_i - \widetilde{c}_i| < \epsilon_i \quad i = 1, 2, 3\}$$

This particular formulation allows us to set different bounds $(\epsilon_1, \epsilon_2, \epsilon_3)$ on the maximum perturbation along each dimension in color space. We also define a constraint based on $\mathcal{F}_{\text{smooth}}$, but instead of using a radius parameter $r$ as in (2) we consider the neighbors $\mathcal{N}(g_j)$ of each lattice point $g_j$ in the grid $\mathcal{G}$:

$$\mathcal{F}_{\text{smooth-col}} \triangleq \{f : \mathcal{C} \to \mathcal{C} \mid \forall g_j \in \mathcal{X}, g_k \in \mathcal{N}(g_j) \quad \|(f(g_j) - g_j) - (f(g_k) - g_k)\|_2 \leq \epsilon_{\text{smooth}}\}$$

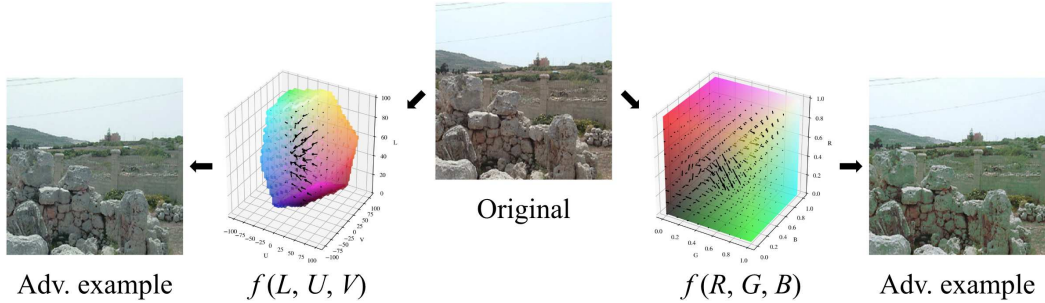

Original

Adv. example     $f(L, U, V)$            $f(R, G, B)$     Adv. example

Figure 4: The color space used affects the adversarial example produced by ReColorAdv. The original image at center is attacked by ReColorAdv with the CIELUV color space (left) and RGB color space (right). The RGB color space results in noticeable bright green artifacts in the adversarial example, while the perceptually accurate CIELUV color space produces a more realistic perturbation.

In the above, $\| \cdot \|_2$ is the $\ell_2$ (Euclidean) norm in the color space $\mathcal{C}$. We define our set of allowable perturbation functions as $\mathcal{F}_{\text{col}} = \mathcal{F}_{\text{diff-col}} \cap \mathcal{F}_{\text{smooth-col}}$ with parameters $(\epsilon_1, \epsilon_2, \epsilon_3, \epsilon_{\text{smooth}})$.

**Optimization**   To generate an adversarial example with ReColorAdv, we wish to minimize $\mathcal{L}_{\text{adv}}(f, x)$ subject to $f \in \mathcal{F}_{\text{col}}$, where $\mathcal{L}_{\text{adv}}$ enforces the goal of generating an adversarial example that is misclassified and is defined as the $f_6$ loss from Carlini and Wagner [3], where $g(\mathbf{x})_i$ represents the classifier's $i$th logit:

$$\mathcal{L}_{\text{adv}}(f, x) = \max \left( \max_{i \neq y}(g(\widetilde{\mathbf{x}})_i - g(\widetilde{\mathbf{x}})_y), 0 \right) \tag{3}$$

When solving this constrained minimization problem, it is easy to constrain $f \in \mathcal{F}_{\text{diff-col}}$ by clipping the perturbation of each color to be within the $\epsilon_i$ bounds. However, it is difficult to enforce $f \in \mathcal{F}_{\text{smooth-col}}$ directly. Thus, we instead solve a Lagrangian relaxation where the smoothness constraint is replaced by an additional regularization term:

$$\underset{f \in \mathcal{F}_{\text{diff-col}}}{\arg\min} \ \mathcal{L}_{\text{adv}}(f, \mathbf{x}) + \lambda \, \mathcal{L}_{\text{smooth}}(f) \tag{$*$}$$

$$\mathcal{L}_{\text{smooth}}(f) \triangleq \sum_{g_j \in \mathcal{G}} \sum_{g_k \in \mathcal{N}(g_j)} \|(f(g_j) - g_j) - (f(g_k) - g_k)\|_2$$

Our $\mathcal{L}_{\text{smooth}}$ is similar to the loss function used by Xiao et al. [27] to ensure a smooth flow field. We use the projected gradient descent (PGD) optimization algorithm to solve $(*)$.

## 4.1   RGB vs. LUV Color Space

Most image classifiers take as input an array of pixels specified in RGB color space, but the RGB color space has two disadvantages. The $\ell_p$ distance between points in RGB color space is weakly correlated with the perceptual difference between the colors they represent. Also, RGB gives no separation between the luma (brightness) and chroma (hue/colorfulness) of a color.

In contrast, the CIELUV color space separates luma from chroma and places colors such that the Euclidean distance between them is roughly equivalent to the perceptual difference [21]. CIELUV presents a color by three components $(L, U, V)$; $L$ is the luma while $U$ and $V$ together define the chroma. We run experiments using both RGB and CIELUV color spaces. CIELUV allows us to regularize the perturbation function $f(\cdot)$ perceptually accurately (see figure 4 and appendix B.1). We experimented with the hue, saturation, value (HSV) and YPbPr color spaces as well; however, neither is perceptually accurate and the HSV transformation from RGB is difficult to differentiate (see appendix C).

Table 1: Accuracy of adversarially trained models against various combinations of attacks on CIFAR-10. Columns correspond to attacks and rows correspond to models trained against a particular attack. C(-RGB) is ReColorAdv using CIELUV (RGB) color space, D is delta attack, and S is StAdv attack. TRADES is the method of Zhang et al. [30]. For classifiers marked (B&W), the images are converted to black-and-white before classification.

| Defense ↓ | None | C-RGB | C | D | S | C+S | C+D | S+D | C+S+D |
|---|---|---|---|---|---|---|---|---|---|
| **Undefended** | 92.2 | 5.9 | 3.0 | **0.0** | 0.9 | 0.8 | **0.0** | **0.0** | **0.0** |
| **C** | 88.7 | 43.5 | 45.8 | 5.7 | 3.6 | 3.4 | 0.9 | **0.2** | **0.2** |
| **D** | 84.8 | 74.9 | 50.6 | 30.6 | 16.0 | 11.7 | 8.9 | 2.7 | **2.2** |
| **S** | 82.7 | 16.9 | 8.0 | 0.5 | 26.2 | 4.8 | **0.0** | 0.1 | **0.0** |
| **C+S** | 89.5 | 31.7 | 23.0 | 0.7 | 10.9 | 8.7 | 0.5 | 0.6 | **0.4** |
| **C+D** | 88.5 | 36.3 | 19.5 | 7.5 | **2.7** | 2.8 | 5.2 | 4.1 | 4.6 |
| **S+D** | 82.1 | 66.9 | 42.7 | 35.4 | 21.9 | 13.4 | 12.2 | 7.6 | **4.1** |
| **C+S+D** | 88.9 | 30.6 | 17.2 | 7.3 | 3.5 | **3.3** | 5.5 | 3.7 | 3.6 |
| **TRADES** | 84.4 | 81.3 | 59.2 | 53.6 | 26.6 | 17.5 | 22.0 | 8.6 | **5.7** |
| **Undefended (B&W)** | 88.3 | 5.3 | 4.1 | **0.0** | 0.9 | 0.6 | **0.0** | **0.0** | **0.0** |
| **C (B&W)** | 85.8 | 40.8 | 38.9 | 4.2 | 2.5 | 2.5 | 0.5 | **0.1** | 0.2 |

## 5    Experiments

We evaluate ReColorAdv against defended and undefended neural networks on CIFAR-10 [13] and ImageNet [20]. For CIFAR-10 we evaluate the attack against a ResNet-32 [6] and for ImageNet we evaluate against an Inception-v4 network [24]. We also consider all combinations of ReColorAdv with *delta* attacks, which use an $\ell_\infty$ additive threat model with bound $\epsilon = 8/255$, and the *StAdv* attack of Xiao et al. [27] that perturbs images spatially through a flow field. See appendix B for a full discussion of the hyperparameters and computing infrastructure used in our experiments. We release our code at `https://github.com/cassidylaidlaw/ReColorAdv`.

### 5.1    Adversarial Training

We first experiment by attacking adversarially trained models with ReColorAdv and other attacks. For each combination of attacks, we adversarially train a ResNet-32 on CIFAR-10 against that particular combination. We attack each of these adversarially trained models with all combinations of attacks. The results of this experiment are shown in the first part of table 1.

**Combination attacks are most powerful**    As expected, combinations of attacks are the strongest against defended and undefended classifiers. In particular, the ReColorAdv + StAdv + delta attack often resulted in the lowest classifier accuracy. The accuracy of only 3.6% after adversarially training against the ReColorAdv + StAdv + delta attack is the lowest we know of.

**Transferability of robustness across perturbation types**    While adversarial attack "transferability" often refers to the ability of attacks to transfer between models [16], here we investigate to what degree a model robust to one type of adversarial perturbation is robust to other types of perturbations, similarly to Kang et al. [11]. To some degree, the perturbations investigated are orthogonal; that is, a model trained against a particular type of perturbation is less effective against others. StAdv is especially separate from the other two attacks; models trained against StAdv attacks are still very vulnerable to ReColorAdv and delta attacks. However, the ReColorAdv and delta attacks allow more transferable robustness between each other. These results are likely due to the fact that both the delta and ReColorAdv attacks operate on a per-pixel basis, whereas the StAdv attack allows spatial movement of features across pixels.

**Effect of color space**    The ReColorAdv attack using CIELUV color space is stronger than that using RGB color space. In addition, the CIELUV color space produces less perceptible perturbations (see figure 4). This highlights the need for using perceptually accurate models of color when designing and defending against adversarial examples.

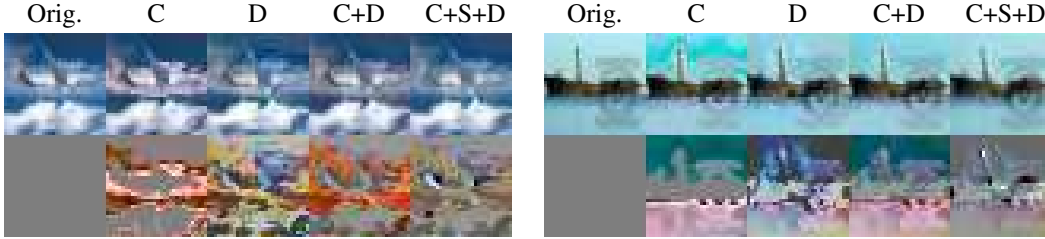

Orig.     C     D     C+D     C+S+D     Orig.     C     D     C+D     C+S+D

Figure 5: Adversarial examples generated with combinations of attacks against a CIFAR-10 WideRes-Net [28] trained using TRADES; the difference from the original is shown below each example. Combinations of attacks tend to produce less perceptible changes than the attacks do separately.

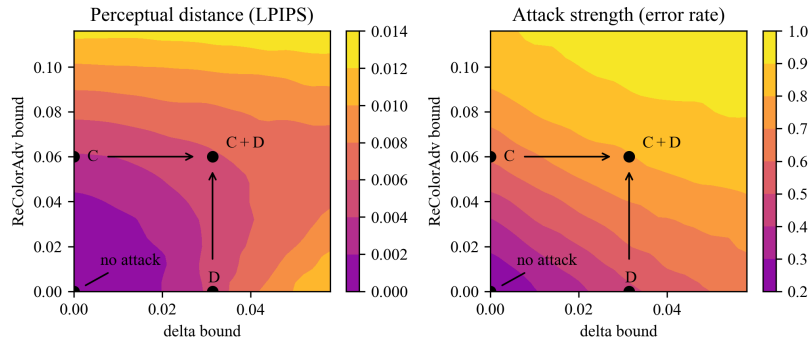

Figure 6: The perceptual distortion (LPIPS) and strength (error rate) of combinations of ReColorAdv and delta attacks with various bounds. The annotated points mark the bounds used in other experiments: C is ReColorAdv, D is a delta attack, and C+D is their combination. Combining the attacks does not increase perceptable change by much (left), but it greatly increases attack strength (right).

## 5.2 Other Defenses

**TRADES**    TRADES is a training algorithm for deep neural networks that aims to improve robustness to adversarial examples by optimizing a surrogate loss [30]. The algorithm is designed around an additive threat model, but we evaluate a TRADES-trained classifier on all combinations of attacks (see the second part of table 1). This is the best defense method against almost all attacks, despite having been trained based on just an additive threat model. However, the combined ReColorAdv + StAdv + delta attack still reduces the accuracy of the classifier to just 5.7%.

**Grayscale conversion**    Since ReColorAdv attacks input images by changing their colors, one might attempt to mitigate the attack by converting all images to grayscale before classification. This could reduce the potential perturbations available to ReColorAdv since altering the chroma of a color would not affect the grayscale image; only changes in luma would. We train models on CIFAR-10 that convert all images to grayscale as a preprocessing step both with and without adversarial training against ReColorAdv. The results of this experiment (see the third part of table 1) show that conversion to grayscale is not a viable defense against ReColorAdv. In fact, the natural accuracy and robustness against almost all attacks decreases when applying grayscale conversion.

## 5.3 Perceptual Distance

We quantify the perceptual distortion caused by ReColorAdv attacks using the Learned Perceptual Image Patch Similarity (LPIPS) metric, a distance measure between images based on deep network activations which has been shown to correlate with human perception [32]. We combine ReColorAdv and delta attacks and vary the bound of each attack (see figure 6). We find that the attacks can be combined without much increase, or with even sometimes a *decrease*, in perceptual difference. As Jordan et al. [10] find for combinations of StAdv and delta attacks, the lowest perceptual difference at a particular attack strength is achieved by a combination of ReColorAdv and delta attacks.

## 6   Conclusion

We have presented *functional threat models* for adversarial examples, which allow large, uniform changes to an input. They can be combined with additive threat models to provably increase the potential perturbations allowed in an adversarial attack. In practice, the ReColorAdv attack, which leverages a functional threat model against image pixel colors, is a strong adversarial attack on image classifiers. It can also be combined with other attacks to produce yet more powerful attacks—even after adversarial training—without a significant increase in perceptual distortion. Besides images, functional adversarial attacks could be designed for audio, text, and other domains. It will be crucial to develop defense methods against these attacks, which encompass a more complete threat model of which potential adversarial examples are imperceptible to humans.

## Acknowledgements

This work was supported in part by NSF award CDS&E:1854532 and award HR00111990077.

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
