[Supplementary Material]

## A  Combining the Additive and Functional Threat Models

Here we provide a proof of Theorem 1.

**Threat model**  Let $\mathbf{x}$ be a grayscale image with $n \geq 2$ pixels, i.e. $\mathbf{x} \in [0,1]^n = \mathcal{X}^n$. Let $t_{\text{add}}$ be an additive threat model where the $\ell_\infty$ distance between input and adversarial example is bounded by $\epsilon_1$, i.e. $\|(\delta_1, \ldots, \delta_n)\|_\infty \leq \epsilon_1$. Let $t_{\text{func}}$ be a functional threat model where $f(x) = c\,x$ for some $c \in [1 - \epsilon_2, 1 + \epsilon_2]$ and let $\epsilon_2 > \epsilon_1 > 0$. The additive threat model allows individually changing each pixel's value by up to $\epsilon_1$; the functional threat model allows darkening or lightening the entire image by up to a proportion of $\epsilon_2$. Both of these are arguably imperceptible perturbations for small enough $\epsilon_1$ and $\epsilon_2$. We also consider $t_{\text{combined}} = t_{\text{add}} \circ t_{\text{func}}$:

$$t_{\text{combined}}(\mathcal{S}) \triangleq \left\{ (c\,x_1 + \delta_1, \ldots, c\,x_n + \delta_n) \left| \begin{array}{l} (x_1, \ldots, x_n) \in \mathcal{S} \\ |\delta_i| \leq \epsilon_1 \\ c \in [1 - \epsilon_2, 1 + \epsilon_2] \end{array} \right. \right\} \tag{3}$$

This combined threat model allows darkening or lightening the image, followed by changing each pixel value individually by a small amount.

**Theorem 1** (restated). *Let $\mathcal{S} \in \mathcal{P}(\mathcal{X}^n)$ be a set of inputs such that $\mathcal{S}$ contains an image that is not too dark; that is, $\exists\, \mathbf{x} \in \mathcal{S}$ for which $\exists\, x_i$ s.t. $x_i > \epsilon_1/\epsilon_2$. Then*

$$t_{\text{combined}}(\mathcal{S}) \;\supsetneq\; t_{\text{add}}(\mathcal{S}) \cup t_{\text{func}}(\mathcal{S}) \qquad \text{or equivalently} \qquad \exists\, \widetilde{\mathbf{x}} \text{ s.t. } \begin{array}{l} \widetilde{\mathbf{x}} \in t_{\text{combined}}(\mathcal{S}) \\ \widetilde{\mathbf{x}} \notin t_{\text{add}}(\mathcal{S}) \cup t_{\text{func}}(\mathcal{S}) \end{array}$$

*Proof.* The above two statements are equivalent, so we focus on the formulation on the right. We calculate $\widetilde{\mathbf{x}}$ and show that it satisfies the given criteria. Let $\mathbf{x} \in \mathcal{S}$ such that $\exists\, x_i$ s.t. $x_i > \epsilon_1/\epsilon_2$. Without loss of generality, assume that in particular $x_2 > \epsilon_1/\epsilon_2$. Then let

$$\widetilde{\mathbf{x}} = ((1 - \epsilon_2)\,x_1 + \epsilon_1, (1 - \epsilon_2)\,x_2, \ldots, (1 - \epsilon_2)\,x_n)$$

First, we show that $\widetilde{\mathbf{x}} \in t_{\text{combined}}(\mathcal{S})$. Using the definition of $t_{\text{combined}}$ in (3), we set $c = 1 - \epsilon_2$, $\delta_1 = \epsilon_1$, and $\delta_2 = \cdots = \delta_n = 0$, which generates $\widetilde{\mathbf{x}}$. These values clearly satisfy the constraints in (3).

Second, we prove that $\widetilde{\mathbf{x}} \notin t_{\text{add}}(\mathcal{S})$ by contradiction. Say that $\widetilde{\mathbf{x}} \in t_{\text{add}}(\mathcal{S})$. Then $\exists\, \delta_1, \delta_2, \ldots, \delta_n$ such that $\widetilde{x}_i = x_i + \delta_i$ and $\|\delta_i\| \leq \epsilon_1$. Consider $\delta_2$, which must satisfy $\widetilde{x}_2 = (1 - \epsilon_2)\,x_2 = x_2 + \delta_2$, or alternatively $\delta_2 = x_2 - (1 - \epsilon_2)\,x_2 = \epsilon_2\,x_2$. However, $x_2 > \epsilon_1/\epsilon_2$ implies that $\delta_2 > \epsilon_1$, which is a contradiction since the constraints on $t_{\text{add}}$ specify that $|\delta_2| \leq \epsilon_1$. Thus, $\widetilde{\mathbf{x}} \notin t_{\text{add}}(\mathcal{S})$.

Third, we prove that $\widetilde{\mathbf{x}} \notin t_{\text{func}}(\mathcal{S})$, again by contradiction. Say that $\widetilde{\mathbf{x}} \in t_{\text{func}}(\mathcal{S})$. Then $\exists\, c \in [1 - \epsilon_2, 1 + \epsilon_2]$ such that $\widetilde{x}_i = c\,x_i$ for all $i$. Considering $i = 1, 2$, we have the following system of equations:

$$\widetilde{x}_1 = cx_1 = (1 - \epsilon_2)\,x_1 + \epsilon_1$$
$$\widetilde{x}_2 = cx_2 = (1 - \epsilon_2)\,x_2$$

From the second equation, we have $c = 1 - \epsilon_2$. However, using this in the first equation gives $(1 - \epsilon_2)\,x_1 = (1 - \epsilon_2)\,x_1 + \epsilon_1$, which implies $0 = \epsilon_1$. This is a contradiction since $\epsilon_1 > 0$, showing that $\widetilde{\mathbf{x}} \notin t_{\text{func}}(\mathcal{S})$. ∎

## B  Experimental Setup

We implement ReColorAdv using the `mister_ed` library [9] and PyTorch [16]. Adversarial examples are generated by 100 iterations of PGD using the Adam optimizer [10] with learning rate 0.001. After all iterations have completed, we choose the result of the iteration with the lowest loss as the adversarial example.

374 When combining attacks, we apply multiple attacks sequentially to the input example and optimize
375 over the parameters of all attacks simultaneously, similarly to Jordan et al. [9].

376 In all adversarial training experiments on CIFAR-10, we begin with a trained ResNet32 [8] and then
377 train it further on batches which are half original training data and half adversarial examples. We
378 adversarially train with a batch size of 500 for 50 epochs. We preprocess images after adversarial
379 perturbation, but before classification, by standardizing them based on the mean and standard
380 deviation of each channel for all images in the dataset. The CIFAR-10 dataset can be obtained from
381 https://www.cs.toronto.edu/~kriz/cifar.html.

382 In CIELUV color space (see section 4.1), we define

$$(c_1, c_2, c_3) = \left( \frac{L}{100}, \frac{U + 100}{200}, \frac{V + 100}{200} \right) \tag{4}$$

383 so that $(c_1, c_2, c_3) \in [0, 1]^3$.

384 For the experiments described in section 5.3, we use LPIPS v0.1 with AlexNet.

## B.1 Regularization Parameters

386 The objective function and constraints described in section 4 include a number of constants that can
387 be used to regularize the outputs of the ReColorAdv attack. Changing these constants alters the
388 strength of the attack and the perceptual similarity of a generated adversarial example to the input.

389 First, $\epsilon_1$, $\epsilon_2$, and $\epsilon_3$ control the maximum amount by which a color in $\mathbf{x}$ can be changed to produce
390 $\widehat{\mathbf{x}}$. For RGB color space, we set $\epsilon_1 = \epsilon_2 = \epsilon_3 = 0.1$; that is, each channel of a color can change by
391 up to $\sim 25/255$. This is greater than the usual $\epsilon = 8/255$ allowed for adversarial examples, but we
392 find that the uniform perturbation used by the functional threat model allows each pixel to change
393 by a greater amount while remaining almost indistinguishable. For the CIELUV color space, we let
394 $\epsilon_1 = \epsilon_2 = \epsilon_3 = 0.06$. This corresponds to a maximum change of 6 in $L$ and a maximum change of 3
395 in $U$ and $V$, since we find that changes in luma are usually less noticeable than changes in chroma.
396 The $\epsilon_i$ values for RGB and CIELUV color spaces result in similar total amounts of perturbation, but
397 the CIELUV color space allows the perturbation to be greater in areas where it is less noticeable.

398 Second, we can control the resolution of the grid $\mathcal{G}$ over which the perturbation function $f(\cdot)$ is
399 parameterized. Let $R_1 \times R_2 \times R_3$ be the resolution of $\mathcal{G}$. Lowering the resolution in a particular
400 dimension acts as a regularizer because it allows less variation in how colors are transformed along
401 that dimension. For RGB color space, we use $R_1 = R_2 = R_3 = 25$. However, for CIELUV color
402 space, we use $R_1 = 16$ and $R_2 = R_3 = 32$. With a high $R_1$ value, we find that the attack sometimes
403 recolors different values of a particular hue very differently. For instance, the attack might make the
404 light parts of a white car green and the dark parts purple. Lowering $R_1$ forces the attack to alter these
405 colors more similarly.

406 Finally, $\lambda$ controls the importance of the smoothness optimization term $\mathcal{L}_{\text{smooth}}$. We always set
407 $\lambda = 0.05$.

## C Learning Rate Experiments

409 We consistently use Adam with a learning rate of 0.001 throughout the main paper to craft adversarial
410 examples. However, we also experimented with a learning rate of 0.01. The results of these
411 experiments are shown below, similar to table 1. All numbers reported are accuracy over the CIFAR-
412 10 test set. Each column corresponds to an attack and each row corresponds to a model trained against
413 a particular attack. C(-RGB) is ReColorAdv using CIELUV (RGB) color space, D is delta attack,
414 and S is StAdv attack. TRADES is the method of Zhang et al. [26]. For classifiers marked (B&W),
415 the images are converted to black-and-white before classification. The learning rate used in an attack
416 is marked above that attack or to the right when the attack is used in adversarial training. There are a
417 couple interesting conclusions that can be drawn from this experiment:

418 • The higher learning rate (0.01) is stronger against TRADES and undefended networks. A
419   ReColorAdv + StAdv + delta (C+S+D) attack with learning rate 0.01 against a TRADES-
420   trained classifier reduces its accuracy to just 6.0%, compared with 10.1% at learning rate
421   0.001.

- Adversarial training against an attack at the higher learning rate (0.01) increases robustness against that attack but lowers it against other attacks. For instance, consider the network defended against C+S+D with learning rate 0.01. This network achieves 15.0% accuracy against attacks of the same type, but the accuracy decreases to 7.1% against some other attacks. In contrast, adversarial training against attacks at the lower learning rate (0.001) leads to more robustness across different attacks.

| Defense | LR | None | C-RGB | C | D | S | C+S | C+D | S+D | C+S+D |
|---|---|---|---|---|---|---|---|---|---|---|
| | | | | | **Attack** (learning rate = 0.01) | | | | | |
| Undefended | | 92.3 | 5.1 | 3.8 | **0.0** | 1.5 | 1.5 | **0.0** | **0.0** | **0.0** |
| C | 0.01 | 87.8 | 37.4 | 45.5 | 4.7 | 3.2 | 2.9 | 1.2 | **0.2** | 0.4 |
| D | 0.01 | 88.8 | 40.4 | 22.7 | 32.7 | 4.2 | 4.3 | 15.0 | 5.0 | **4.0** |
| S | 0.01 | 89.3 | 11.5 | 9.8 | 0.3 | 29.0 | 9.5 | 0.4 | 0.4 | **0.3** |
| C+S | 0.01 | 90.5 | 27.0 | 24.3 | 2.8 | 31.4 | 23.0 | **2.1** | 2.8 | **2.1** |
| C+D | 0.01 | 88.3 | 46.3 | 32.7 | 34.4 | 6.0 | 5.4 | 22.6 | **4.7** | 4.8 |
| S+D | 0.01 | 88.0 | 25.3 | 17.4 | 28.4 | 9.3 | **8.1** | 22.6 | 17.0 | 13.9 |
| C+S+D | 0.01 | 89.0 | 32.3 | 23.8 | 29.8 | 13.4 | **11.4** | 26.1 | 17.4 | 15.0 |
| C | 0.001 | 89.2 | 37.2 | 46.6 | 5.1 | 3.4 | 3.0 | 1.1 | **0.3** | **0.3** |
| D | 0.001 | 84.7 | 72.9 | 57.4 | 30.8 | 12.2 | 11.2 | 12.6 | 2.4 | **1.8** |
| S | 0.001 | 82.7 | 14.9 | 11.9 | 0.5 | 22.2 | 6.7 | **0.1** | 0.2 | **0.1** |
| C+S | 0.001 | 82.3 | 37.5 | 40.4 | 5.9 | 18.5 | 13.1 | 1.9 | 0.9 | **0.8** |
| C+D | 0.001 | 84.3 | 70.8 | 60.0 | 33.8 | 9.4 | 8.7 | 18.1 | **1.8** | 1.9 |
| S+D | 0.001 | 82.0 | 65.2 | 49.9 | 35.0 | 18.5 | 14.0 | 16.5 | 5.5 | **4.5** |
| C+S+D | 0.001 | 82.3 | 65.8 | 53.0 | 34.8 | 16.8 | 14.7 | 18.3 | 5.1 | **5.0** |
| TRADES | | 84.2 | 79.7 | 69.2 | 53.5 | 21.0 | 17.8 | 33.8 | 6.6 | **6.0** |
| Undefended (B&W) | | 87.9 | 4.7 | 4.8 | **0.0** | 1.6 | 1.5 | **0.0** | 0.1 | **0.0** |
| C (B&W) | 0.01 | 84.7 | 40.4 | 41.7 | 4.5 | 2.4 | 2.4 | 1.0 | **0.2** | 0.3 |
| C (B&W) | 0.001 | 85.6 | 37.8 | 40.7 | 4.0 | 2.5 | 2.5 | 0.7 | **0.3** | **0.3** |
| | | | | | **Attack** (learning rate = 0.001) | | | | | |
| Undefended | | 92.3 | 8.3 | 5.3 | **0.0** | 2.2 | 1.8 | **0.0** | **0.0** | **0.0** |
| C | 0.01 | 87.8 | 46.2 | 48.4 | 5.9 | 4.5 | 4.4 | 1.6 | **0.3** | 0.7 |
| D | 0.01 | 88.8 | 43.7 | 25.4 | 26.4 | 4.1 | **3.8** | 15.7 | 8.3 | 7.9 |
| S | 0.01 | 89.3 | 18.7 | 13.9 | 0.4 | 13.8 | 8.4 | 0.8 | **0.6** | 0.9 |
| C+S | 0.01 | 90.5 | 39.1 | 32.3 | 4.3 | 22.7 | 17.5 | **2.8** | 3.5 | 3.3 |
| C+D | 0.01 | 88.3 | 49.1 | 35.6 | 29.2 | **5.3** | 5.4 | 20.3 | 8.7 | 8.3 |
| S+D | 0.01 | 88.0 | 25.8 | 17.7 | 10.7 | 4.5 | **4.1** | 9.3 | 6.1 | 5.9 |
| C+S+D | 0.01 | 89.0 | 33.9 | 24.9 | 15.7 | 7.5 | **7.1** | 13.0 | 8.4 | 8.5 |
| C | 0.001 | 89.2 | 47.4 | 50.3 | 5.9 | 4.6 | 4.6 | 1.7 | **0.5** | 0.9 |
| D | 0.001 | 84.7 | 77.3 | 61.9 | 32.8 | 18.6 | 17.2 | 17.3 | 4.3 | **4.2** |
| S | 0.001 | 82.7 | 20.3 | 15.7 | 0.8 | 29.9 | 10.7 | **0.2** | **0.2** | **0.2** |
| C+S | 0.001 | 82.3 | 47.2 | 44.6 | 7.5 | 26.2 | 20.2 | 3.5 | 2.2 | **2.0** |
| C+D | 0.001 | 84.3 | 74.7 | 63.5 | 35.2 | 14.0 | 13.4 | 22.2 | 4.5 | **4.2** |
| S+D | 0.001 | 82.0 | 69.3 | 53.9 | 36.5 | 26.4 | 21.1 | 21.7 | 9.6 | **8.0** |
| C+S+D | 0.001 | 82.3 | 70.1 | 56.4 | 35.5 | 25.5 | 21.4 | 23.4 | 10.0 | **8.5** |
| TRADES | | 84.2 | 81.6 | 72.8 | 53.7 | 31.2 | 27.5 | 39.3 | 11.1 | **10.1** |
| Undefended (B&W) | | 87.9 | 7.3 | 6.1 | **0.0** | 1.6 | 1.6 | **0.0** | **0.0** | **0.0** |
| C (B&W) | 0.01 | 84.7 | 49.3 | 44.9 | 5.4 | 4.1 | 3.8 | 1.6 | 0.5 | 0.7 |
| C (B&W) | 0.001 | 85.6 | 46.7 | 43.8 | 5.0 | 3.5 | 3.8 | 1.3 | **0.4** | 0.7 |

# D Non-Additive Threat Models

Here, we discuss some other non-additive adversarial threat models that have been explored in the literature and how our work differs from them.

**Spatial Threat Models**    Some recent work has focused on *spatial threat models*, which allow for slight perturbations of the locations of features in an input rather than perturbations of the features themselves. Xiao et al. [23] propose StAdv, which optimizes the parameters of a smooth flow field that moves each pixel of an input image by a small, bounded distance to generate an example that fools the classifier. Wong et al. [22] bound the Wasserstein distance between the original input and the adversarial example. Engstrom et al. [3] apply an small rotation and translation to an input image to generate a misclassification.

**Other Threat Models**    A few papers have focused on threat models that are neither additive or spatial. Zeng et al. [25] perturb the properties of a 3D renderer to render an image of an object which is unrecognizable to a classifier or other machine learning algorithm. Hosseini and Poovendran [6] propose "Semantic Adversarial Examples," which allow modifications of the input image's hue and saturation. Hosseini et al. [7] also explore inverting images to cause misclassification. These latter two papers can be considered as special examples of functional threat models. In the first, each pixel's hue and saturation is shifted by the same amount; that is, each pixel is transformed by the function $f(h, s, v) = (h + \delta_h, s + \delta_s, v)$. In the second, each pixel is inverted, i.e. each pixel channel is transformed by the function $f(x_i) = 1 - x_i$. However, the authors do not propose a general framework for these types of attacks, as we do. Furthermore, the adversarial examples generated by these attacks are often not realistic and not imperceptible. For example, their crafted adversarial examples include green skies, purple fields of grass, and inverted street signs—unlike our proposed ReColorAdv attack, which results in imperceptible changes.

# E Additional Images

Figure 7: More adversarial examples like those in figure 2, generated by ReColorAdv against an Inception-v4 classifier on ImageNet. Top row: original images; middle row: adversarial examples; bottom row: magnified difference.

Figure 8: More adversarial examples like those in figure 5, generated with combinations of attacks against a CIFAR-10 WideResNet trained using TRADES. C is ReColorAdv, D is delta attack, and S is StAdv attack [23]. The difference from the original is shown to the right of each example. Combinations of attacks tend to produce less perceptible changes than the attacks do separately.

## F  Lipschitz Regularization

In addition to the regularizations defined in section 3.1, we can also enforce that the perturbation function $f(\cdot)$ in a functional threat model is Lipschitz for some suitably small $\kappa$:

$$\mathcal{F}_{\text{lips}} \triangleq \{f : \mathcal{X} \to \mathcal{X} \mid \forall x_1, x_2 \in \mathcal{X} \; \|f(x_1) - f(x_2)\| \leq \kappa \|x_1 - x_2\|\} \tag{5}$$

$\mathcal{F}_{\text{lips}}$ requires some smoothness in the perturbation function $f(\cdot)$, ensuring that similar features in the input are mapped to similar features in the adversarial example. However, one disadvantage of $\mathcal{F}_{\text{lips}}$ is that it includes constant functions $f(x) = c$, i.e. functions which map every feature to a single value, removing salient features from the input. Thus, we ultimately use $\mathcal{F}_{\text{smooth}}$ instead.