[Reviews · NeurIPS 2019]

Reviewer 1



#####After Rebuttal##### I think the authors to clarify my concerns. For the second question, I would like to see more instances of functional attacks, which could be also applied to image classification. Given such results would make this paper stronger. I will keep my original rating. ##################### This paper proposes a novel class of threat models for crafting adversarial examples. The paper is well-written. A typical type of functional adversarial attacks is realized by changing the color of images as ReColorAdv. The constrains on this attack and the optimization process is clearly illustrated. Below are some minor concerns: 1. Although the proposed functional adversarial attack is novel, it is somewhat relevant to "blind-spot attack" (Zhang et al., "The Limitations of Adversarial Training and the Blind-Spot Attack", ICLR 2019), and "unrestricted adversarial examples" (Song et al., "Constructing Unrestricted Adversarial Examples with Generative Models", NeurIPS 2018). The authors can discuss the connections between the proposed attack and other attacks, and also the differences. 2. Although the definition of functional threat model is general and flexible, this paper only provides one instance of functional attacks. More examples of functional attacks can make this paper more convincing and interesting. 3. From Table 1, C and C-RGB attacks are less powerful than L_\infty attack (D). S+D attack is more powerful than C+D in most cases, and gets similar performance to C+S+D. So the concern is about the effectiveness of the proposed ReColorAdv attack. It seems that ReColorAdv brings little benefit.

Reviewer 2



Update after authors response: Authors addressed my concerns regarding strength of the attack in their response. ---------------------------- Original comments: Originality: Authors propose novel adversarial attack which recolors all pixels of the image in the same way using function f(x). Authors also propose to combine multiple adversarial attack to build stronger adversary. Quality: Overall paper is technically sound. One piece of critics is following. It seems like proposed attack is actually much weaker compared to other attacks (as could be seen from table 1). However this could be compensated by the fact that proposed attack seems to be less noticeable visually and also could be combined with other attacks. Clarity: Paper is clearly written. Significance: Moderate significance. Authors propose interesting new technique for crafting adversarial perturbations which in most cases visually less noticeable compared to L-infinity ball restricted perturbations.

Reviewer 3



This paper would be more interesting if it could do more than just adjust the brightness on each channel independently. Would he approach work taking a tripple of values (or ) and perturb those functionally also be effective using this definition? How much more power would that have? The proposed approach is interesting. New threat models are generally useful and help expand the space of valid attacks. I was especially happy to see a discussion around the color space. (However: I was missing a definition of CIELUV --- was it defined anywhere?) It would have been nice to see more discussion around this point, and in particular setting different bounds per channel. For example, it's well studied that humans are more sensitive to changes in green than blue (and so standard compression will compress blue more than green). Can the same be modeled here? My main concern with this paper is it is not technically deep: there is not much novelty gained on top of the basic idea. However, it is evaluated well and clearly presented. There are no major writing issues. Two minor comments: - I am relatively well-versed in the literature, but even I don't know what function "f6" is from Carlini & Wagner [2]. - In 5.1 using the word "transferability" is somewhat confusing as this is often the terminology used for adversarial examples. Reply to Author Response: Thank you for the clarifications and experiments. I am increasing my score as a result.

[Author Response · NeurIPS 2019]

1 We thank the reviewers for their insightful feedback.

2 **Strength of attack** Some of the reviewers raised concerns about the strength of the ReColorAdv attack relative to existing attacks. We perform two new experiments to demonstrate the strength of ReColorAdv. First, we found an implementation issue with the projection step of PGD for non-RGB color spaces in the original paper; we fixed it and found that the strength of the attack increased. In addition, we evaluated all attacks with 300 iterations of PGD. The results are shown in the table below. Note that the combined ReColorAdv + StAdv + delta attack reduces the accuracy of an adversarially trained classifier to just 3.6%, less than half of the previous best combined attack. Finally, the success rate of the ReColorAdv attack is not its only advantage; the perturbations it produces are less noticeable as well.

| Defense | PGD iters | Attack | | | | | | |
|---------|-----------|--------|------|------|------|------|------|-------|
|         |           | C      | D    | S    | C+S  | C+D  | S+D  | C+S+D |
| None | 100 | 4.4 | 0.0 | 2.2 | 1.6 | **0.0** | **0.0** | **0.0** |
| None | 300 | 3.3 | 0.0 | 1.2 | 0.9 | **0.0** | **0.0** | **0.0** |
| Adv. training | 100 | 50.2 | 32.8 | 29.9 | 15.4 | 14.9 | 9.6 | **8.8** |
| Adv. training | 300 | 45.8 | 30.1 | 26.2 | 8.7 | 5.2 | 7.6 | **3.6** |
| TRADES | 100 | 64.8 | 53.7 | 31.2 | 23.2 | 29.0 | 11.1 | **8.1** |
| TRADES | 300 | 59.2 | 53.6 | 26.6 | 17.5 | 22.0 | 8.7 | **5.7** |

10 **Related work (R1)** Like our work, both Song et al.[1] and Zhang et al.[2] aim to construct adversarial examples outside of the usual $\ell_p$ ball perturbation. However, Song et al. uses a generative model to craft adversarial examples directly without perturbing other images; in contrast, we aim to augment the space of adversarial *perturbations* of images in the train or test sets. Zhang et al. is more similar to our work; they apply a single affine function to all pixels in an input before performing PGD. However, unlike our work, they do not consider more complex functions and they do not present their "scale and shift" technique as an attack in itself, only as a way of discovering images far from the training manifold. We will incorporate a discussion of similarities and differences to these works in the revised draft.

17 **Other functional attacks (R1)** The functional adversarial threat model is applicable to many domains but we choose to focus on image classification. This problem is widely viewed as a benchmark in machine learning and adversarial robustness research and focusing on it allows us to report in-depth results. While we hope to extend the threat model to other domains in the future, we believe it is beyond the scope of this paper.

21 **Perturbations of triples vs. separate channels (R3)** Indeed ReColorAdv does operate on triples rather than perturbing each channel independently, i.e. each pixel is mapped $(R, G, B) \to f(R, G, B)$.

23 **Other color spaces (R3)** Upon your suggestion, we experimented with the HSV (hue, saturation, value) and YPbPr color spaces in addition to RGB and CIELUV. HSV presents difficulties when performing PGD because the derivative of the transformation from RGB is highly discontinuous; thus we use an approximation HSV' which maps colors into a hexagonal pyramid instead of the standard HSV cone. A disadvantage of both HSV and YPbPr is that they were originally designed for transmitting video signals rather than as an accurate representation of how humans view colors.

28 Below are adversarial examples based on suggestions from R3, which we will also include in the paper. In these experiments, we present ReColorAdv using four color spaces; see C-{LUV, RGB, YPbPr, HSV'}. We also apply ReColorAdv separately to each channel. i.e. $(R, G, B) \to (f_1(R), f_2(G), f_3(B))$; see C-{LUV, RGB}-Sep-Channels. Finally, we use separate bounds on each RGB channel based on the sensitivity of the human eye ($\epsilon_R = 0.1, \epsilon_G = 0.05, \epsilon_B = 0.15$); see C-RGB-Sep-Bounds. Note that we already applied separate bounds in CIELUV color space, as detailed in appendix B.1. For each of these variations, the accuracy of an undefended model under the attack is shown.

| Original | C-LUV | C-LUV-Sep-Channels | C-RGB | C-RGB-Sep-Channels | C-RGB-Sep-Bounds | C-YPbPr | C-HSV' |
|----------|-------|--------------------|-------|--------------------|------------------|---------|--------|
| 92.3% | 4.4% | 8.4% | 8.2% | 9.3% | 8.4% | 2.6% | 2.1% |

36 **Minor comments (R3)** We appreciate these suggestions and will fix them in the revised draft.

[1] Yang Song et al. Constructing Unrestricted Adversarial Examples with Generative Models. *NeurIPS*, 2018.
[2] Huan Zhang et al. The Limitations of Adversarial Training and the Blind-Spot Attack. *ICLR*, 2019.


[Meta-Review · NeurIPS 2019]

This paper proposed functional adversarial attack, by applying the same transform on each of the input features (e.g. pixels) in order to fool the classifier. As a practical algorithm, the authors proposed ReColorAdv which changes the colors of an input image uniformly. The authors discussed a constrained optimisation approach to find the functional adversarial example, and proposed to combine different threat models (additive and functional) to construct stronger attacks. Experiments are performed on CIFAR-10 and the proposed method successfully fools a ResNet classifier with defense methods such as adversarial training and TRADES. All the reviewers are experts in designing adversarial attacks. They found the proposed approach novel overall, but initially they were concerned about the strength of the attack. This concern is addressed in the author feedback. I would suggest the following revision for camera ready: 1. Add in the extra experiments provided in the author feedback, to demonstrate further the strength of the attack; 2. If possible, discuss possible defense against the proposed attack, this will help the readers to understand how strong the attack is; 3. Discuss in detail the difference between the proposed approach and existing approaches (e.g. unconstrained adversarial examples).